# Maximum Likelihood Instead of Least Squares in Fracture Analysis by Means of a Simple Excel Sheet with VBA Macro

Vincenzo Guerriero 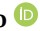

Department of Civil, Construction-Architectural and Environmental Engineering (DICEAA), University of L'Aquila, 67100 L'Aquila, Italy; vincenzo.guerriero@univaq.it

**Abstract:** This technical note illustrates a linear regression algorithm based on the Maximum Likelihood Estimation (MLE), with a related Excel spreadsheet and VBA program, adapted to the case of fracture aperture data sets in which sampling of the smallest values is problematic. The method has been tested by means of Monte Carlo simulations and exhibits significantly better convergence against Least Squares criterion (LSM). As the method is conceptually simple and, following the indications illustrated here, the relative spreadsheet can be easily designed, it may be routinely used, instead of the Least Squares, in fracture analysis. Furthermore, the proposed method, with the appropriate modifications, might be potentially extended to other cases in geology and geophysics, in which significant biases at the lower limits of the sampling scale occur.

**Keywords:** Maximum Likelihood; fracture statistical analysis; Monte Carlo method; MS Excel; visual basic for applications





## 1. Introduction

Stratabound joints, as defined by Odling et al. [1] (also defined as perfect bed-bounded joints [2]), are fractures exhibiting in outcrop both terminations on the bed boundaries (Figure 1a). Let us consider the fracture sub-network composed only of this kind of joint. Such joint system, together with bedding joints bounding the mechanical layers, can form a well-connected hydraulic network [3], observable at the meter–decimeter scale, which divides rock into blocks that are parallelepiped shaped, often clearly observable in field.

In the last decades several studies (e.g., [4–7]) have highlighted how permeable structures are normally detectable on several scales of observation. Other studies have emphasized the advantages of using multiple-porosity models to simulate hydraulic behavior of naturally fractured rocks [8–14]. Multiple-porosity models allow to consider the different permeable structures in rock (e.g., pores, vugs, fracture networks, from micro-fractures to faults, observable at different scales) as hydraulic systems that are conventionally distinct, overlapping and interacting among them (e.g., [15] and references therein).

Guerriero et al. [4] proposed a hierarchical model for permeable structures in carbonate rocks, which identified four main kinds of interacting hydraulic structures, observable at different scales: (i) fault network, (ii) stratabound joints, (iii) non-stratabound joints (including micro- fractures) and (iv) a pore scale system. Stratabound joint networks assume a key role in the hydraulic behavior of fractured rock masses because they are an element of communication between the small-scale fractures (including micro-fractures) with the large-scale fault network.

For these reasons, the geometric characterization of stratabound joint networks in terms of fracture aperture and spacing may be of considerable interest. In fact, for a given succession, the knowledge of how aperture and spacing vary with bed thickness, would allow, for a given succession, a hydraulic characterization or modeling of stratabound joint networks, based on thickness data. Joint spacing and its dependence on rock layer thickness has been extensively analyzed [1,16–23]. Joint aperture and related statistical

distribution have also been studied [2,4,6,24–32]. Nevertheless, in the specific case of perfect bed-bounded (i.e., stratabound) joints, aperture statistical distribution and its dependence on bed thickness and/or lithology has been insufficiently studied.

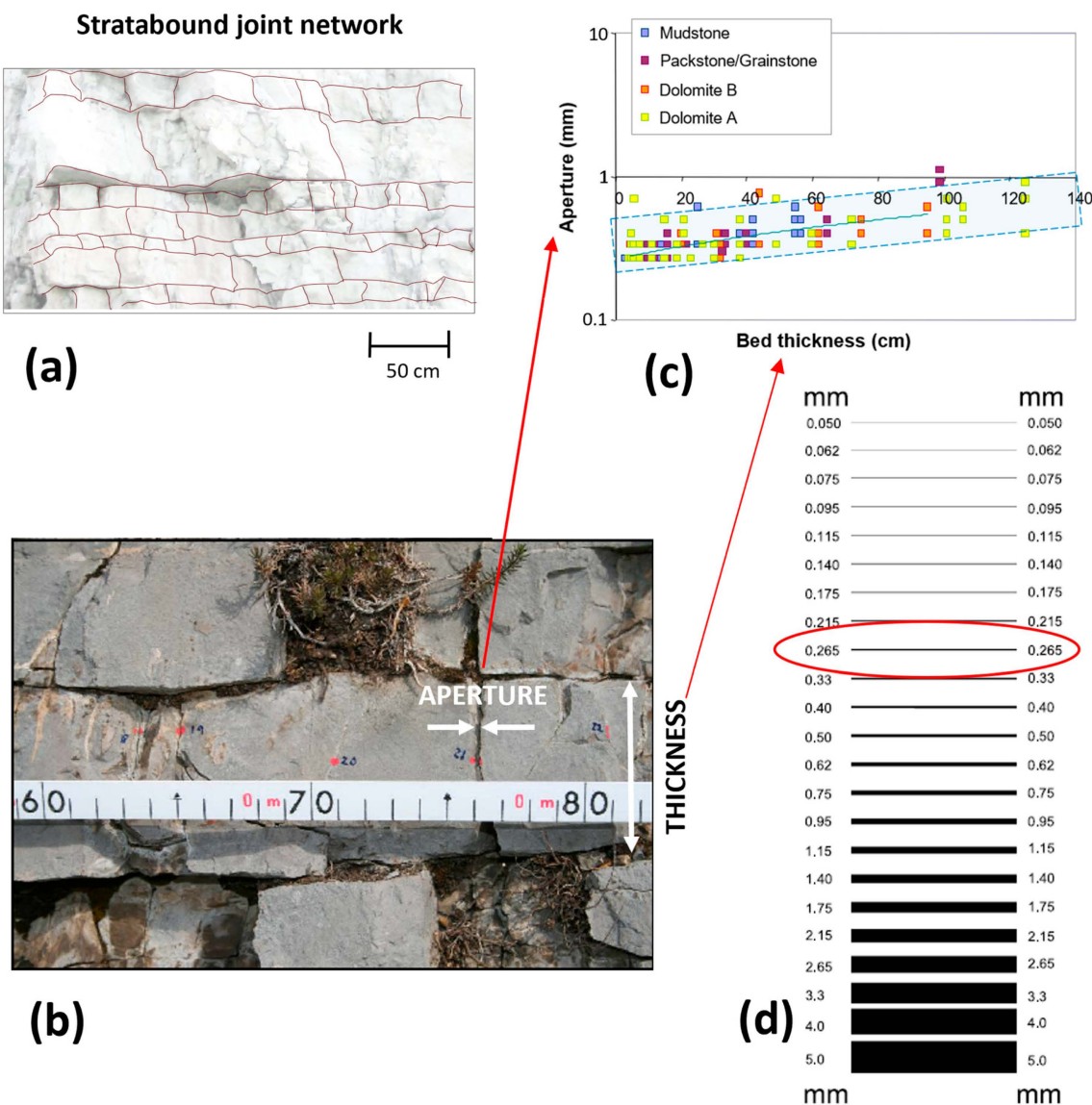

**Figure 1.** (**a**) Stratabound or perfect bed-bounded joint network; (**b**) example of field scanline; (**c**) example of field data, provided as paired values of joint aperture vs. bed thickness, highlighted by red arrows (from [3], modified); (**d**) comparator utilized for field measuring of joint aperture (from [6], modified); the threshold value of 0.265 mm is highlighted by a red circle.

Therefore, Guerriero et al. [3] carried out a specific statistical investigation aimed at assessing whether and how stratabound joint aperture depends on bed thickness and/or lithology. Such analysis involved two Lower Cretaceous (Albian) carbonate successions, outcropping at Faito and Chianello Mts. (southern Italy). These successions, which had been selected as surface analogues of buried oil reservoirs in Val D'Agri (Basilicata, Italy), were previously well studied in terms of sedimentology and petrophysics [33,34], as well as of geological structural settings [3,30,31,35].

The statistical analysis carried out by Guerriero et al. [3] pointed out that stratabound joint aperture depends on bed thickness, according to a linear function, characterized by a non-zero y-intercept. This may have important consequences for the hydraulic and structural characterization of fractured rocks. As explained by Guerriero et al. [3], non-

zero intercept implies that packages of thin beds may exhibit very high porosity and permeability values, whereas thicker beds are expected to exhibit low porosity but high permeability values.

In that analysis, fracture sampling was carried out by measuring aperture and bed thickness for each joint (and other attributes that are not relevant for the present study), along scanlines oriented parallel to the bedding along a vertical face through the center of each fracture (Figure 1b,c). To measure joint aperture in the field, the caliper proposed by Ortega et al. [6] was utilized (Figure 1d).

Guerriero et al. [3] explained the various drawbacks associated with field measurement of fracture aperture. Based on the visibility conditions at the studied outcrops, aperture measurements that were less than 0.265 mm (Figure 1d) were considered unreliable. For this reason, it was decided to set a minimum threshold value of 0.265 mm for them, thus including all aperture values detected equal to or less than this limit into this single category. This produces a data truncation that should not be confused with the truncation artifact (e.g., [6]) because the latter is an underestimation of the number of fractures detected, and small fractures (and micro-fractures) might not be visible. In the case of stratabound joints in our study area, we expect that they are practically always visible (as these do not include micro-fractures), but it is the measure of their aperture that could be wrong. By way of example, if in a certain layer six joints are measured with the following (true) aperture values (mm) [0.316, 0.224, 0.118, 0.081, 0.426, 0.277], the values of the recorded measurements according to the above-mentioned scale and threshold are [0.33, 0.265, 0.265, 0.265, 0.4, 0.265]. If the layer has, e.g., a thickness of 28 cm, the sample consists of the following pairs of values [(0.33, 28), (0.265, 28), (0.265, 28), (0.265, 28), (0.4, 28), (0.265, 28)]. Therefore, the number of fractures sampled is not altered (i.e., no truncation artifact, sensu Ortega et al., [6]), but there is a loss of information about the recorded measurement values.

The inclusion of all smaller aperture values into a single aperture class may induce a marked tendency of the regression line, in a diagram showing aperture vs. bed thickness, to intersect the ordinate axis near this threshold [3]. Because the intercept value identified by Guerriero et al. [3] was just close to this limit, the suspicion arose that it could have been affected by an artifact. Therefore, it has been decided to repeat the analysis of those data using a more effective method based on Maximum Likelihood Estimation (MLE). Such criterion has been successfully used by Rizzo et al. [32] to study fracture aperture and length statistics, which demonstrated the validity of the MLEs against linear regression. In the present work, the MLE has been used for different purposes, i.e., for the specific goal of correcting the above-mentioned aperture measurement bias within the framework of linear regression analysis.

The aim of this paper is to describe the adopted statistical method and the utilized algorithm and software, as well as to provide details about its validation and effectiveness. To this end, an example of application based on synthetic simulated fracture data is illustrated to clarify the differences between the two approaches and the added value of the Maximum Likelihood approach. Instead, illustrating the results of the analysis involving real field data from Faito and Chianello outcrops [3] goes beyond the scope of this work and these will be presented in a companion paper to be published later.

## 2. Recalls about Maximum Likelihood Estimation in Linear Regression

Let us consider a certain observed statistical sample and assume that it follows a known probability distribution. MLE is a parametric method in which the parameters (e.g., mean, variance, etc.) are estimated by adjusting them until the probability of observing that sample is maximized (e.g., [36]). As an example, suppose we observe an event that we know to be distributed according to a Poisson distribution, which occurs on average $\lambda$ times in a time interval T. Let us imagine that we have the following outcomes of six independent observations (i.e., that we have observed the occurrences for six different time intervals

T): [0, 3, 5, 6, 3, 5]. If we knew the $\lambda$ parameter, we could calculate the probability $L(\lambda)$ associated with this sample as the product of the probabilities of each observation:

$$L(\lambda) = \frac{\lambda^0 \cdot e^{-\lambda}}{0!} \cdot \frac{\lambda^3 \cdot e^{-\lambda}}{3!} \cdot \frac{\lambda^5 \cdot e^{-\lambda}}{5!} \cdot \frac{\lambda^6 \cdot e^{-\lambda}}{6!} \cdot \frac{\lambda^3 \cdot e^{-\lambda}}{3!} \cdot \frac{\lambda^5 \cdot e^{-\lambda}}{5!} \ . \tag{1}$$

In case $\lambda$ is not known and we want to estimate it, the basic idea of MLE consists of finding the one that maximizes $L(\lambda)$ among all the possible values of $\lambda$, as seen in Equation (1). The function $L(\lambda)$ is called Likelihood Function (LF). This function is also used for continuous probability distributions as the product of probability density functions. By way of example, if we have N observations of a Normal variable z, as $[Z_1, Z_2, \ldots Z_N]$, then the LF is given by:

$$L(\mu, \sigma) = \frac{1}{\sigma\sqrt{2\pi}} \prod_{j=1}^{N} e^{-\frac{(z_j - \mu)^2}{2\sigma^2}}; \tag{2}$$

where $\mu$ and $\sigma$ denote the parameters mean and standard deviation, respectively. In the case where $\mu$ and $\sigma$ are both unknown, LF must be minimized as a function of two variables. If one of the two parameters is known, then LF must be minimized with respect to the other unknown parameter. The cases above illustrated that the parameter values which maximize the LF can be calculated in closed form. If the involved equations are more complicated, maximization can be carried out numerically. Usually, the logarithm of the LF, denoted by Log-Likely Function, is utilized (e.g., [36,37]). In case of numerical maximization, its use has the advantage of avoiding overflow/underflow drawbacks.

The MLE can also be employed in (linear or nonlinear) regression analysis, as an alternative method to LSM [37,38]. In regression analysis between the two variables z and T, it is assumed to be:

$$z_i = y(T_i) + r_i, \tag{3}$$

where $y(T_i)$ (i.e., the expected value) is a function of T, and r is a random residual with zero mean and constant standard deviation. In linear regression, the expected value y is defined as a linear function of T: y = m T + n, where m and n are constant parameters. If the analysis is carried out by means of MLE (which is a parametric method), a kind of distribution needs to be assumed for r. Usually, r is assumed normally distributed, with zero mean and constant standard deviation (here denoted by $\sigma$). Considering Equation (2), the LF involving r is:

$$L = \frac{1}{\sigma\sqrt{2\pi}} \prod_{j=1}^{N} e^{-\frac{r_j^2}{2\sigma^2}} = \frac{1}{\sigma\sqrt{2\pi}} \prod_{j=1}^{N} e^{-\frac{(z_j - y(T_j))^2}{2\sigma^2}} = \frac{1}{\sigma\sqrt{2\pi}} \prod_{j=1}^{N} e^{-\frac{(z_j - (m \cdot T_j + n))^2}{2\sigma^2}}. \tag{4}$$

It should be noted that, in Equation (4), $z_j$ and $T_j$ are known observed values, whereas m, n and $\sigma$ need to be calculated by maximizing $L(m,n,\sigma)$ or its logarithm.

It can be proven that such an approach to linear regression provides the same results as the classical LSM (e.g., [37]). Nevertheless, in the present study, the LF is defined in a different way (Section 3.1), and Section 3.2 explains why such MLE formulation provides different results than LSM.

Guerriero et al. [3] suggested that it is more appropriate to calculate residuals $r_i$ as the difference between the logarithms of observed values $z_i$ and predicted $y_i$:

$$r_i = \ln(z_i) - \ln(y_i). \tag{5}$$

Also in this case, residual r is assumed normally distributed, with zero mean and constant standard deviation denoted by d. Furthermore, the expected value is again: y = m T + n. It should be noted that this definition implies that, for each rock layer, i.e., fixed T, the result is $\ln(z) = r + \ln(y(T))$; therefore, $\ln(z)$ is the sum of a Normal variable with zero mean (i.e., r) and a constant (i.e., $\ln(y(T))$), i.e., it is a Normal variable with a mean equal to such constant. In other words, within each layer, the observed aperture is well

described by a Log Normal variable (i.e., whose logarithm is Normal), with a median value equal to y(T) = m T + n, and standard deviation d.

Once r, z and y are defined, an expression of the LF similar to that given in Equation (4) may be found. In the present study, a differently defined LF is used, whose details are illustrated in the next section.

## 3. Methods

### 3.1. Linear Regression by Means of Maximum Likelihood Estimation

Here, the LF involved in regression analysis, and utilized by the proposed algorithm and spreadsheet, is defined. Given the joint aperture classes illustrated in Figure 1d, let us denote the limit value by $x_i$ between contiguous classes $s_i$. In this instance, $x_i$ is an intermediate value between $s_{i-1}$ and $s_i$, opportunely chosen (Figure 2, at #1). Under the hypothesis that joint aperture values exhibit Log Normal distribution (Sect. 2), let us denote a probability distribution by $F_{mn·d}(x)$ whose mean is ln($y$) and the standard deviation is $d$. Here, $y$ denotes the expected aperture value, which is a linear function of bed thickness $T$, whose parameters are coefficient $m$ (mm/cm) and intercept $n$ (mm). Then, the probability that a measured aperture value $S$ falls within the class $s_i$ denoted by $p_{mn,d}(s_i)$, for i > 1, is:

$$\text{Probability}(x_{i-1} < S < x_i) = p_{mn,d}(s_i) = F_{mn,d}(x_{i-1}) - F_{mn,d}(x_i); \quad i > 1; \quad (6)$$

Whilst, for the first aperture class:

$$\text{Probability}(S < x_1) = p_{mn,d}(s_1) = F_{mn,d}(x_1); \quad i = 1. \quad (7)$$

Therefore, for a given sample [$S_k$, $T_k$], the Likelihood Function $L(m,n,d)$ assumes the following form:

$$L(m, n, d) = \prod_k p_{mn,d}(S_k). \quad (8)$$

Searching for the maximum of this function on the space of the three parameters $m$, $n$ and $d$ (numerically), the MLE estimates of these three parameters are achieved.

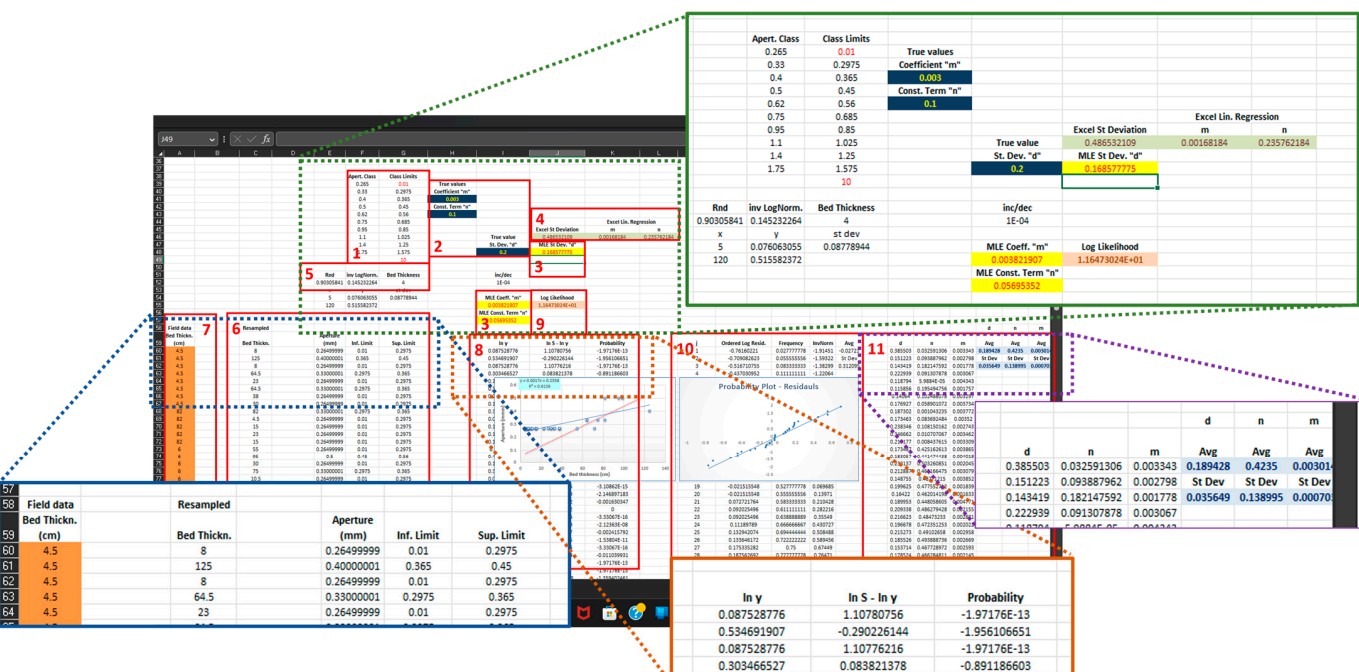

**Figure 2.** Spreadsheet utilized for simulations. Yellow cells with red font denote variables to be adjusted in order to maximize the object function. Dark blue cells with yellow font denote parameter true values. #1: Aperture classes and related limits. #2: True *m*, *n* and *d* values of the model simulating

fracture data. #3: Estimated *m*, *n* and *d* values, by MLE. #4: Estimated *m*, *n* and *d* values, by Excel linear regression functions. #5: Excel function LOGINV() to produce a single random aperture value, starting from a bed thickness value. #6: Simulated data set; from left: resampled bed thickness value, fracture aperture class and its upper and lower limits. #7: Field bed thickness data. #8: Likelihood data (see main text). #9: Object function; in sheet *MLE*, it is the Log Likelihood, and in sheet *LSM*, it is the sum of square of residuals. #10: Data to build up probability plots of residuals. #11: Monte Carlo simulation output data. Three columns on the left include each one 100 estimated values of *m*, *n* and *d*; on the right side, average and standard deviation of these columns are calculated.

### 3.2. Different Response to Data Truncation of the Proposed MLE and LSM, in Linear Regression

To understand the systematic error in the LSM analysis, caused by data truncation, let us recall the example illustrated in the Introduction, in which we imagine having acquired six joint aperture measurements from a 28-cm-thick layer. Imagine reporting these values in a diagram with bed thickness in abscissae and joint aperture in ordinate (e.g., Figure 1c). Of the six values above illustrated, three of them (with true apertures of 0.224, 0.118 and 0.081 mm) should have been placed in aperture classes lower than 0.265 mm. The grouping of these aperture values in the single class of 0.265 mm produces an upward migration of the points associated with the related measure. If we imagine inserting data recorded from several layers into this diagram, such migration of points will occur more frequently for smaller thicknesses (Figure 1c). This migration produces a lifting of the left part of the trendline, resulting in an increase in the value of the intercept, with consequent alteration of the estimated coefficient, due to a clockwise rotation of the trendline.

Whenever an experimental point has ordinates in the 0.265 mm class, the LSM introduces a bias due to the assumption that the residual around the trendline can take on the values +r and −r with equal probability. Due to data truncation, the measured aperture values can take on any exceeding values (according to the scale in Figure 1d) but cannot take on values lower than 0.265 mm. The MLE, on the other hand, does not introduce this bias, since the definition of the LF (Equation (8)) associates with values belonging to the 0.265 mm aperture class, and the probability that an aperture value is equal to or lower than this threshold (true statement). If a sample contains many aperture values in that class, LSM may provide highly biased estimates, while MLE is expected to be unaffected by that condition.

### 3.3. The Excel Sheet and VBA Program

The Excel folder utilized in this work includes three sheets: *MLE*, *LSM* and *Results*. Figure 2 illustrates the sheet *MLE* in detail; with respect to the latter, the sheet *LSM* is different only in column K cells, whose formula calculate the square of residual in the adjacent cell in column J. The first 30 rows, which are not depicted here, include a header illustrating some user instructions. The routines, written in Visual Basic for Applications (VBA), which utilize this folder to analyze data and carry out Monte Carlo simulations, are:

- *Sub Maximize()* and *Sub Minimize_LS():* analyze a data set by maximizing or minimizing an object function, which is Log likelihood for the former and residual standard deviation for the latter.
- *Sub Simul_Apert_Data():* based on thickness field data (#7 in Figure 2) and assigned (true) values of the parameters *m, n* and *d* (#2 in Figure 2), it produces a data set, composed of 35 paired values, by (i) resampling thickness data and (ii) producing a random aperture value for each thickness value (Section 3.3). Then, it identifies which class, and related limits, belongs to (#6).
- *Sub Simul_100*(): for each triplet of true values *m, n* and *d*, it produces 100 simulated data sets and analyzes each one by means of *Sub Maximize()*, in sheet MLE, and *Sub Minimize_LS()*, in sheet LSM. Then, it saves the estimated values in columns S, T and U.

- *Sub Monte_Carlo():* varies the *n* true value in the range 0.05–4.75 mm, and for each value produces simulations by calling *Sub Simul_100()*, then saves the related results (#11) in sheet *Results* (Figure 3).

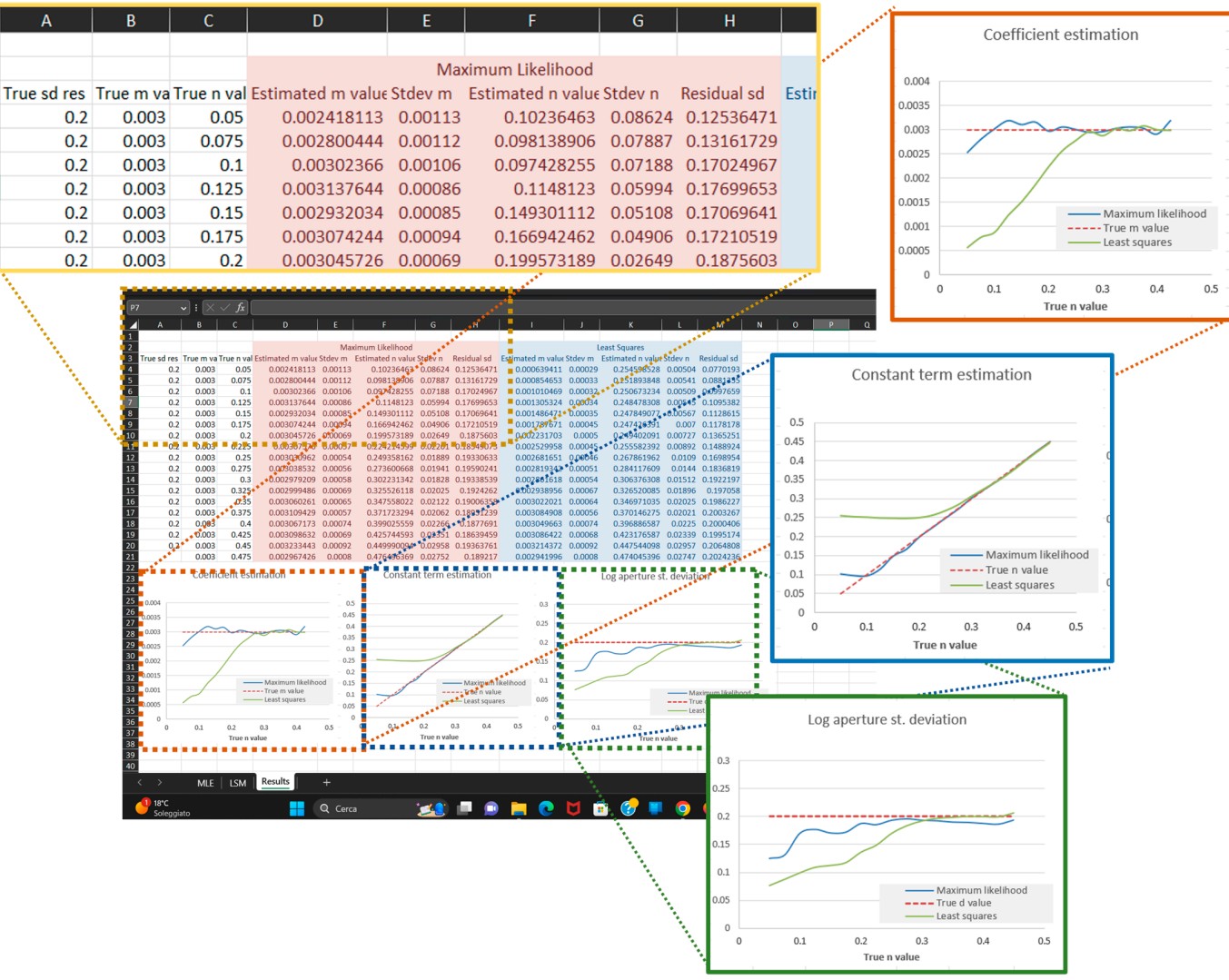

**Figure 3.** Spreadsheet "Results". The diagrams on the left side point out that MLE estimators exhibit modest deviations over the whole analyzed range of true *n* values and excellent convergence for *n* > 0.1, whereas LSM ones converge only for *n* > 0.3, showing large deviations elsewhere.

The core of the calculation method in the *MLE* sheet is in formulas in column *K* (#8 in Figure 2), in which for each aperture value (in column *E*, #6), the probability logarithm that it falls within the range to which it belongs is calculated as follows:

$$\text{LN}(\text{LOGNORMDIST}(G60; \text{LN}(I\$55 \times C60 + I\$57); J\$48) - \text{LOGNORMDIST}(F60; \text{LN}(I\$55 \times C60 + I\$57); J\$48)$$

The formula $\text{LOGNORMDIST}(G60; \text{LN}(I\$55 \times C60 + I\$57); J\$48)$ provides the probability that an aperture value is lesser or equal to the limit in cell G60 when its median value is a linear function of thickness (term $LN(I\$55 \times C60 + I\$57)$) and its standard deviation is the value d in cell J48. The difference between distribution values in the formula above illustrated the probability that an aperture value lies within the range limited by cells G60–F60. The sum of logarithms of these probabilities returns the Log likelihood in cell J55. The routine Sub Maximize() starts calculations by assigning likely initial values to

parameters *m*, *n* and *d* achieved by Excel least squares functions in cells J46:L46 (#4) by means of instructions such as:

$$Range("I55") = Range("K46")$$

$$Range("I57") = Range("L46")$$

$$Range("J48") = Range("J46")$$

then, it iteratively adjusts the values of parameters *m*, *n* and *d* in cells I55, I57 and J48, respectively (#3, in Figure 2), according to a simple steeper descent algorithm, until the maximum of the Log Likelihood (cell J55) is reached.

The spreadsheet also includes formulas to build up probability plots of residuals, indicated in panel #10 in Figure 2. Illustrating the use of these plots lies outside the scope of this paper. An explanation of their construction and usefulness is provided, such as in the work of Chambers et al. [39].

### 3.4. Validation by Means of Monte Carlo Simulation

A convenient way to evaluate the effectiveness of the MLE method compared to the LS consists of carrying out the analysis on simulated data sets whose parameters are already known, then comparing the known parameter values with the estimates achieved according to the two methodologies. To this end, a series of Monte Carlo simulations was carried out in which synthetic fracture aperture data sets were produced using likely values for the parameters m and d (equal to 0.03 and 0.2, respectively) and with parameter *n* (which was critical in our analysis) varying in the range 0.050–0.475 mm according to the following procedure:

1.  Produce a data set, constituted by paired values (aperture, thickness), using the known parameters (*m,n,d*),
2.  Simulate data truncation, including all joints belonging to the aperture class of 0.265 mm or lesser, in the 0.265 mm class,
3.  Analyze by means of MLE,
4.  Analyze by means of LS,
5.  Compare known parameter values with estimates by MLE and LS,
6.  Go back to step #1.

From sheet *MLE*, the *Sub Monte_Carlo()* is called, which assigns, as "true" values from which simulated data are produced, *m* = 0.003, *d* = 0.2 and varies *n*, starting from 0.050 to 0.475 with step of 0.025. For each *n* value, 35 thickness values are chosen from field data (in column A) by means of random numbers (*Sub Simul_Apert_Data()*). For each thickness value, an aperture one is produced by means of a random number and of the function in cell F52:

$$LOGINV(E52; LN(H41 \times G52 + H43); I\$48)$$

which returns the inverse Log Normal distribution of: (1) random number, (2) mean as logarithm of a linear function with *m* value from cell H41 and *n* from H43 and (3) standard deviation from cell I48. Then, the subroutine *Sub Simul_Apert_Data()* individuates the class and related limits to which this value belongs, and saves these in columns E, F and G. Hence, the *Sub Maximize()* is called, which maximizes the Log Likelihood. Then, the simulated data set is copied within the sheet *LSM*, and the best fit line is calculated by minimizing the sum of the square of residuals, calculated according to Equation (5) (*Sub Minimize_LS()*).

For each *n* value, this procedure is repeated 100 times; then, (*m*, *n*, *d*) estimates are saved in columns U, T and S, respectively. After 100 iterations, the average and standard deviation of these column values are saved into the sheet *Results*. Then, *n* is incremented by 0.025 mm.

## 4. Use of the Spreadsheet for Analysis of Field Data

The illustrated spreadsheet allows us to reproduce the mentioned Monte Carlo simulations. Nevertheless, it can be used to analyze our own field data by means of the routines *Sub Initialize_Apert_Data()* and *Sub Maximize()*. The fracture aperture data will be entered in column E and, for each item, the associated bed thickness value will be entered at the same row, in column C. Then, call the routine *Sub Initialize_Apert_Data()*, which will insert the appropriate values into columns F and G by individuating the class and related limits to which each aperture value belongs. Then, call the *Sub Maximize()* on the MLE sheet, which will estimate the parameters m, n and d, by maximizing the object function (i.e., the Log Likelihood function). This routine only works on the sheet denoted by MLE.

The spreadsheet, in its current form, analyzes the data set of 35 items. When analyzing a data set of a different size—for example, 50 items—the arrays in columns I, J, K, M, N, O and P need to be updated so that they have the same size (i.e., 50). Furthermore, the formulas in cells J55, J46, K46 and L46 must be updated so that they perform calculations on arrays of suitable length. By way of example, the formula in cell J55, which calculates the Log Likelihood, is:

$$\text{SUM(K60:K94)}.$$

This calculates the Log Likelihood as sum of items in column K and rows from 60 to 94. In cases in which, for example, the analyzed data set includes 50 values, then the last one will fall at row 109. Therefore, such a formula will need to be updated as SUM(K60:K109).

This spreadsheet utilizes the aperture classes according to the logarithmic comparator proposed by Ortega et al. [6], which are stored in column F, at rows from 39 to 48. The class limits are in column G. By way of example, the limit between the aperture class of 0.33 mm (in cell F40) and 0.4 mm (in cell F41) is given by their average, equal to 0.365 mm (in cell G41). In case it is needed to use different aperture classes, in addition to modifying the values in that column, the routine *Sub Initialize_Apert_Data()* also needs to be updated in order to take into account both the varied limit values for each class and the varied number of aperture classes.

## 5. Results Discussion

Figure 3 shows the results of Monte Carlo simulations aimed at comparing the MLE and modified LSM results. The estimates of the parameters m, n and d, are plotted against the true n value, here denoted by $n_{true}$. The MLE estimates of m and n exhibit a good convergence over the whole analyzed $n_{true}$ range. The estimates of the three parameters m, n and d, achieved by MLE, show an excellent convergence for $n_{true} > 0.1$ mm. The LSM estimates of m, n and d show notably strong deviations from true values, for $n_{true} < 0.2$ mm (i.e., LSM provides strongly biased estimators), and a good convergence only for $n_{true} > 0.3$ mm. It is noteworthy that, for $n_{true} < 0.2$ mm, the LS estimate of the parameter n assumes the same value (about 0.25 mm) and is independent of the statistical sample. This highlights the absolute ineffectiveness of such a statistical approach in estimating this parameter for small $n_{true}$ values.

Table 1 illustrates the results of Monte Carlo simulations for the MLE, which are partly visible in Figure 3. For each parameter analyzed, the estimate and, in the adjacent column, the relative standard deviation are reported. For $n_{true} > 0.1$ mm, the mean values of m, n and d estimates converge, thus pointing out that MLE provides unbiased estimators. The standard deviation of the parameter n estimate increases as $n_{true}$ decreases. This highlights a random uncertainty that can be reduced by increasing the statistical sample size (as the estimator is unbiased).

**Table 1.** Results of Monte Carlo simulations. For each parameter, the mean value of the sampling estimate is illustrated, as well as the related standard deviation, as a descriptor of uncertainty. Note that the 6th column shows the estimated value of the parameter 'residual standard deviation', whereas the last column shows the related standard deviation (i.e., the uncertainty).

| | Method: Maximum Likelihood Estimation True m = 0.003, True Residual std dev = 0.2, Sample Number = 35 | | | | | |
|---|---|---|---|---|---|---|
| **True n Value** | **Estimated m** | **Std dev. m** | **Estimated n** | **Std dev. n** | **Estimated Residual std dev.** | **Std dev. Residual std dev.** |
| 0.05 | 0.0024 | 0.00113 | 0.102 | 0.086 | 0.125 | 0.105 |
| 0.075 | 0.0028 | 0.00112 | 0.098 | 0.079 | 0.132 | 0.089 |
| 0.1 | 0.0030 | 0.00106 | 0.097 | 0.072 | 0.170 | 0.088 |
| 0.125 | 0.0031 | 0.00086 | 0.115 | 0.060 | 0.177 | 0.085 |
| 0.15 | 0.0029 | 0.00085 | 0.149 | 0.051 | 0.171 | 0.071 |
| 0.175 | 0.0031 | 0.00094 | 0.167 | 0.049 | 0.172 | 0.053 |
| 0.2 | 0.0030 | 0.00069 | 0.200 | 0.026 | 0.188 | 0.052 |
| 0.225 | 0.0031 | 0.00057 | 0.224 | 0.022 | 0.185 | 0.047 |
| 0.25 | 0.0030 | 0.00054 | 0.249 | 0.019 | 0.193 | 0.043 |
| 0.275 | 0.0030 | 0.00056 | 0.274 | 0.019 | 0.196 | 0.039 |
| 0.3 | 0.0030 | 0.00058 | 0.302 | 0.018 | 0.193 | 0.035 |
| 0.325 | 0.0030 | 0.00069 | 0.326 | 0.020 | 0.192 | 0.034 |
| 0.35 | 0.0031 | 0.00065 | 0.348 | 0.021 | 0.190 | 0.033 |
| 0.375 | 0.0031 | 0.00057 | 0.372 | 0.021 | 0.190 | 0.032 |
| 0.4 | 0.0031 | 0.00074 | 0.399 | 0.023 | 0.188 | 0.031 |
| 0.425 | 0.0031 | 0.00069 | 0.426 | 0.024 | 0.186 | 0.030 |
| 0.45 | 0.0032 | 0.00092 | 0.450 | 0.030 | 0.194 | 0.029 |
| 0.475 | 0.0030 | 0.00080 | 0.476 | 0.028 | 0.189 | 0.027 |
| 0.05 | 0.0024 | 0.00113 | 0.102 | 0.086 | 0.125 | 0.105 |

## 6. Concluding Remarks

MLE is particularly effective in analyzing fracture data sets in which the field measurement of minor fracture aperture is problematic. Comparison of MLE against LSM has pointed out that the former is able to remove the bias due to data truncation, and then provide unbiased estimators, whereas the latter provides strongly biased estimators. The utilized algorithm, the related spreadsheet and routines have been illustrated in detail, thus allowing the reader to use the Excel folder proposed here and modify it or to create their own version. As this linear regression method can be easily performed with an Excel spreadsheet, it may be routinely used in fracture analysis. Furthermore, with the appropriate modifications, it might be potentially extended to other experimental situations where biases occur at the lower limit of the sampling scale.

**Funding:** Ministry of Economic Development (MiSE), Italy, Grant Id: C19C20000520004.

**Data Availability Statement:** Name of the code/library: MonteCarlo_MLE.xlsm. Contact: vincenzo.guerriero@univaq.it or vincenzo.guerriero@unina.it. Program language: Visual Basic, Applications Edition (VBA). Software required: MS Office or equivalent. Program size: 82 KB. The source codes are available for downloading at the link: https://github.com/vincenzo-guerriero/MonteCarlo_MLE.git (accessed on 2 December 2023).

**Acknowledgments:** The author thanks the editor and anonymous reviewers for their constructive comments and suggestions, which helped to significantly improve this manuscript. The research leading to these results has received funding from the Italian Ministry of Economic Development (MiSE) under the project "SICURA—CASA INTELLIGENTE DELLE TECNOLOGIE PER LA SICUREZZA CUP. C19C20000520004—Piano di investimenti per la diffusione della banda ultra larga FSC 2014-2020".

**Conflicts of Interest:** The author declares no conflict of interest.

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
