# Peer review of "Maximum Likelihood Instead of Least Squares in Fracture Analysis by Means of a Simple Excel Sheet with VBA Macro"

_geosciences, doi:10.3390/geosciences13120379_

Round 1

Reviewer 1 Report

Comments and Suggestions for Authors

The paper "Maximum Likelihood Instead of Least Squares in Fracture

Analysis by Means of a Simple Excel Sheet with VBA Macro" presents a simple and practical algorithm, along with an Excel sheet and VBA program, for analyzing fracture aperture data sets with problematic sampling of smallest values. The exposition of this work needs to be substantially improved. The authors superficially explain various vital points. Several contents must be appropriately discussed, with the due background. Also, the presentation is sometimes confusing, with missing or poorly motivated definitions. The introduction presents the concepts of fracture analysis well but nothing about maximum likelihood and least squares methods. The methodology should be better and more detailed presented. The findings need to be better presented. Also, the equations are poorly written and with little discussion around them. Some important equations should have been included. Numerical experiments should be better described. It is recommended that the authors revise your paper to address these issues adequately; providing more information about the data, possible comparisons with other models, and discussing the potential bias in the approaches would significantly strengthen your paper. Please see some comments below. 

(1) The authors maintain that maximum likelihood estimation is used instead of the least squares approach in fracture analysis to reduce possible biases related to sampling small fractures. The authors must clarify this point in the new version of the manuscript. 

(2) The least squares approach is based on maximum likelihood estimation based on the Gaussian distribution. The authors show that considering a log-normal distribution via the maximum likelihood method is more appropriate. In fact, considering a log-normal distribution means carrying out a least squares process in which the difference between predicted and observed values is given by Eq. (1). In other words, the authors are actually comparing least squares with least squares, only changing the values of z to ln(z). Such discussion must be included in the new version of the manuscript. 

(3) The authors should discuss the maximum likelihood method and its relationship with least squares problems in the introduction section. This discussion is critical to providing a solid foundation for the results presented and enhancing the reader's understanding. To fill this gap, I suggest including pertinent references, such as https://doi.org/10.1016/S0022-2496(02)00028-7 [provides a concise explanation of maximum likelihood estimation framework as a robust estimation method] and https://doi.org/10.1140/epjp/s13360-021-01521-w [shows the relationship between (generalized) least squares estimates and the maximum likelihood approach], which can enrich the discussion and the robustness of our work. These sources offer valuable insights into applying the maximum likelihood method in least squares contexts and will complement this manuscript.

(4) The authors performed a visual inspection of the results, but it is recommended to include at least two error measures for a more complete and objective comparison between the results depicted in Figure 2. Visual analysis is valuable, but quantitative measures will provide a more solid basis for evaluating the effectiveness of the methodologies in comparison. Thus, it is recommended to consider widely recognized error measures such as root-mean-square error (RMSE) and/or mean absolute error (MAE) and coefficients of determination (R²) such as the Pearson linear correlation. Incorporating these error/similarity metrics will enrich the analysis.

(5) The caption for Figure 1 is too long. Reduce it and describe this figure in the body of the manuscript.

(6) Lines 131-15 have no meaning.

(7) Section 4 should be separated into two sections, one with a detailed discussion of the results (and its presentation) and the next presenting the final remarks.

Comments on the Quality of English Language

Proofreading is recommended.

Reviewer 2 Report

Comments and Suggestions for Authors

Line 115, 158: Figure 1 does not look clear. Please increase the DPI.

Reviewer 3 Report

Comments and Suggestions for Authors

The article deals with the implementation of a Maximum Likelihood method for fracture analysis of soils. The methodology has been implemented in an excel sheet and compared with Least Squares method to highlight its strength points. The paper is well structured and fits with the journal scope. The English language require some minor correction. However, some consideration must be considered before publication in Geosciences.

1.     Abstract, line 9: change “was problematic” with “is problematic”.

2.     Page 2, line 59: change “ruler” with “rule” or “approach”.

3.     Page 3, line 106: what is “Pr”? Please, describe it in the text.

4.     Figure 1 caption: it is too long. Please, describe the content of Figure 1 within the text.

5.     Page 4, line 141: it seems that something is missing within the parenthesis.

6.     Page 5, lines 165-166: insert the expression as an equation to be more readable.

7.     Page 6, line 202: same consideration of comment 6.

Finally, I’m really concerned about the rate of auto citation of this article. On 24 references, 10 of them are of the author, reaching an auto citation rate of 41%. I recommend to discus in the introduction other work related in the same field. Please, do a more precise review of the literature.

For all the previous reasons, the reviewer recommends minor amendments of paper for publication in Geosciences.

Comments on the Quality of English Language

The English language require some minor correction.

Round 2

Reviewer 1 Report

Comments and Suggestions for Authors

The author addressed all issues raised in the first round of reviews.

Reviewer 3 Report

Comments and Suggestions for Authors

Dear Author, thank you for the effort in improving the paper. You have addressed all the questions in a clear way. The quality of the paper has been improved, so, I reccomend to accept the paper in the current form for publication in Geosciences